# IP6 Regulation of HIV Capsid Assembly, Stability, and Uncoating

**DOI:** 10.3390/v10110640

**Published:** 2018-11-15

**Authors:** Robert A. Dick, Donna L. Mallery, Volker M. Vogt, Leo C. James

**Affiliations:** 1Department of Molecular Biology and Genetics, Cornell University, Ithaca, NY 14853-2703, USA; vmv1@cornell.edu; 2Protein and Nucleic Acid Chemistry Division, Medical Research Council, Laboratory of Molecular Biology, Cambridge CB2 0QH, UK; dlm@mrc-lmb.cam.ac.uk

**Keywords:** HIV, IP6, capsid, infection, uncoating, AIDS

## Abstract

The mechanisms that drive formation of the HIV capsid, first as an immature particle and then as a mature protein shell, remain incompletely understood. Recent discoveries of positively-charged rings in the immature and mature protein hexamer subunits that comprise them and their binding to the cellular metabolite inositol hexakisphosphate (IP6) have stimulated exciting new hypotheses. In this paper, we discuss how data from multiple structural and biochemical approaches are revealing potential roles for IP6 in the HIV-1 replication cycle from assembly to uncoating.

## 1. Introduction

The HIV structural protein Gag and its products are involved at multiple steps in the viral replication cycle. The first step, assembly, takes place during the formation and budding of virus particles from an infected cell. The multi-domain structural protein Gag (Figure 1A) polymerizes into an immature lattice underneath and contacts the plasma membrane. During or after release of the virus by budding, the viral protease PR then cleaves Gag into its constituent domains and leads to a structural rearrangement called maturation. The liberated CA protein, which comprises two separately folded sub-domains (CA_NTD_ and CA_CTD_), forms a new lattice known as the mature capsid that surrounds the RNA genome and associated proteins including reverse transcriptase (RT). Both the immature (Gag) and the mature (CA) lattice are built of hexamers, but the molecular intra-hexamer and inter-hexamer contacts differ. In addition, the mature CA lattice contains 12 pentamers that are required to ensure that the lattice and resulting capsid are completely closed. The mature capsid is deposited into the cytoplasm of a target cell following viral envelope fusion with the cell membrane. There the capsid protects the virus from sensors and degradative enzymes, ensures efficient reverse transcription of the viral genome from RNA to DNA, and facilitates transport through the cytosol and into the nucleus. The precise interplay of these events is unclear but requires capsid interaction with a number of host co-factor molecules. The final part of this step is uncoating, i.e. the partial or complete dissolution of the capsid lattice. Exactly where, when, and how uncoating happens is hotly debated, but it is likely more complex than the immediate and spontaneous collapse pictured in early models. In this review, we will consider how the recently identified co-factor molecule IP6 impacts these steps.

## 2. Assembly

As for most other retroviruses, HIV-1 Gag oligomerizes at the plasma membrane to create a curved lattice that makes up the protein shell of the immature virus particle [1]. The assembly properties of Gag have been studied extensively in vitro, starting some 20 years ago [2,3,4]. In this system, purified Gag protein or Gag truncation mutants are mixed with a nucleic acid, which is typically single-stranded DNA or RNA that interacts with the highly basic NC domain and, thus, brings protein molecules in close proximity. Under the right conditions, rapid, spontaneous assembly occurs, which results in the formation of spherical or tubular/conical virus-like particles (VLPs) closely resembling virus particles that have budded from cells. For HIV-1, immature assembly proceeds most favorably at slightly basic pH and physiological ionic strength. By contrast, the CA protein by itself and the CA-NC protein in the presence of nucleic acid assemble into the mature lattice, which is a reaction that is more efficient at a slightly acid pH. Purified CA protein, which does not require addition of nucleic acid, forms a mature lattice only at unphysiologically high salt such as 1 M NaCl. The products of CA and CA-NC assembly are mostly tubes, which are composed of hexamers, in addition to some closed core-like structures that are composed of hexamers and 12 pentamers [4,5,6].

A truncated HIV-1 Gag protein missing most of the MA domain (delta 16–99) forms spherical particles with an immature lattice, which was shown over 15 years ago [7]. However, under the conditions used initially, full-length Gag, which contains an intact (but not myristoylated) MA domain, forms only aberrant small spherical particles in vitro [2]. This size defect could be corrected by the addition of a small amount of a rabbit reticulocyte extract to the assembly reaction. The active ingredient in the extract was identified as inositol pentakisphosphate or IP5 [8] with inositol hexakisphosphate (IP6 or phytic acid) later shown to be equally active. IP6 is found in mammalian cells at concentrations in the range between 12 to 100 µM and is the most abundant of the several multiphosphorylated inositols [9]. The biosynthetic pathway leading to this metabolite is complex and proceeds via a phosphorylation cascade from IP3 [10].

How these inositol phosphate compounds function to correct immature assembly was the topic of several earlier studies. One used a foot printing technique to demonstrate that in vitro IP6 interacts with multiple Lys residues in Gag with the strongest being in the MA and NC domains [11]. A subsequent study showed that IP6 could induce assembly of Gag in the absence of nucleic acid when the NC domain was replaced with a Leucine Zipper domain, which is a construct that forms VLPs in vivo [12]. HIV-1 MA interacts with the plasma membrane (PM) through hydrophobic and electrostatic interactions. In particular, MA interacts specifically with the phospholipid PI(4,5)P2 [13,14]. This minor phospholipid is required for effective Gag targeting to the PM for assembly and budding [15,16]. As a purified protein, HIV-1 Gag adopts a horseshoe-like conformation with the N-terminal MA domain in proximity to the C-terminal NC domain [17,18]. IP6 was found to induce Gag to take on an elongated, assembly-competent conformation [19]. Taken together, these several observations suggested that IP6 acts on the MA domain of Gag.

In a recent paper, Dick et al. provided molecular evidence for the mechanism of IP6 action in promoting immature assembly [20]. They showed first that the MA domain is not required for this effect. The protein CA-SP-NC assembles inefficiently and yields mature tubes and immature spheres. However, in the presence of physiological concentrations of IP6, CA-SP-NC assembly is increased by as much as 100-fold with the major product being immature spherical VLPs. Thus, IP6 must bind either to the CA, SP, or NC domains. To further delineate the site of interaction, the ability of IP6 to promote assembly of CA-SP was tested. Remarkably, IP6 also causes this highly truncated Gag protein to form immature VLPs, ruling out the NC domain as the primary site of action. Our findings do not rule out that IP6 interaction with MA and NC plays a role in Gag assembly.

A feature of Gag that plays a key role in immature assembly is a stretch of 14 amino acids spanning the CA_CTD_ and SP interface (Figure 1A). This region, SP1, forms a six-helix bundle (6HB) that stabilizes immature Gag hexamers [21,22] and is required for immature lattice formation [7]. The six negatively charged phosphate groups of IP6 roughly define a ring of diameter ~10 Å. Since a Gag hexamer is the basic building block of the immature lattice, Dick et al. reasoned that a ring of positively charged residues within the Gag hexamer could create a binding pocket with an arginine (R) or lysine (K) side chain contacting each phosphate. Furthermore, a 4.5 Å cryo-EM structure of immature HIV-1 virus particles [22] as well as a 3.27 Å crystal structure of a CA_CTD_-SP hexamer [21] show two K residues, K290 at the bottom of the CA_CTD_ hexamer and K359, near the top of the 6HB. Although the side-chains are not visible in these structures, each lysine could potentially form a ring with the diameter of a size that could accommodate an IP6 molecule. These two rings would be stacked roughly on top of each other and in a cryo-EM structure obtained from enveloped immature virus particles that budded from cells, an undefined density feature of the correct size for IP6 is observed between and inside the two positively charged rings [22]. Lastly, hexacarboxybenzene, which is a compound structurally similar to IP6 with six negatively charged groups, was previously shown to bind to an analogous ring of positively charged residues at the center of the mature CA hexamers [23].

Consistent with the hypothesis that a ring of charged residues in the CA_CTD_-SP binds IP6, IP6 was shown to promote formation of microcrystals by this construct with the crystals having a hexameric lattice, which was observed by 2D cryo-EM [20]. Higher resolution X-ray crystallography of these crystals directly demonstrated that IP6 sits in a pocket formed by the two rings of six K residues with direct interactions between the IP6 phosphates and the side chains of both K290 and K359 (Figure 2A). All-atom molecular dynamic simulations of CA_CTD_-SP with and without IP6 further support the conclusion that IP6 acts to stabilize the six-helix bundle. Other inositol derivatives and hexacarboxybenzene also stabilize the 6HB in this computational model, which is consistent with their stimulation of immature assembly in vitro. Most importantly, mutation of either of K290 or K359 to alanine results in a dramatic decrease in the efficiency of in vitro assembly.

Does IP6 function in living cells to promote HIV-1 Gag assembly? Dick et al. using media collected from virus-producing cells transfected with K290A or K359A plasmids observed no infections [20]. CRISPR/Cas9-mediated knockout of inositol-pentakisphosphate 2-kinase (IPPK), which is an enzyme involved in the biosynthesis of IP6, led to more than a 10-fold decrease in HIV-1 infections [20]. These results are consistent with a role for IP6 in either viral production or infection. A prediction of the hypothesis that IP6 promotes physiological HIV-1 immature assembly is that IP6 will be incorporated into infectious viral particles. Virion incorporation experiments have previously been used to show packaging of important cofactors (e.g., cyclophilin A [24]) and antiviral restriction factors (e.g., APOBEC3G [25]). By producing HIV-1 in cells supplemented with tritiated inositol, Mallery et al. demonstrated that IP6 is incorporated into the virus at a level of about 300 molecules per virion [26]. IP6 is selectively enriched over IP4 and IP5, which suggests that, while other inositol derivatives can promote assembly in vitro, IP6 is the physiologically relevant molecule. The 300 molecules of IP6 are sufficient for most but not all of the immature hexamers in an HIV-1 virus particle based on the estimated stoichiometry of roughly 2500 Gag molecules per virus particle depending on the level of incompleteness of the Gag shell [27]. Qualitatively, this is consistent with the finding that a substoichiometric ratio of IP6 to hexamer is sufficient to promote efficient in vitro assembly.

## 3. Maturation

HIV-1 virions that have budded from cells undergo maturation. During this step, the viral protease cuts Gag in several places including in the 6HB, which likely disrupts the immature lattice and allows the dissociation of IP6. The resulting CA protein then assembles into the mature conical capsid core. Previous work had demonstrated that, at the center of each mature capsid hexamer, is a ring of six arginine residues (R18) that avidly binds polyanions including the IP6-like hexacarboxybenzene [23]. More recent X-ray structures confirm that the R18 ring in the mature hexamer can also bind IP6 [20,26] (Figure 2B). Evidence from cryo-EM tomographic analysis also suggests that a small molecule is bound at this site in mature virions [28]. It is, therefore, an attractive hypothesis that IP6 might drive assembly of the mature capsid inside virions in a similar manner to immature lattice formation. Supporting this hypothesis, Dick et al. demonstrated that IP6 promotes the assembly of recombinant CA protein into a mature lattice. As described above, it is only at unphysiologically high ionic strength that HIV-1 CA had been observed to form tubular and some conical structures in vitro. The lattice of these products is representative of the lattice of cores in budding virus particles from cells [28]. However, the addition of IP6 to a CA assembly reaction at physiological ionic strength results in massive and unprecedented formation of core-like structures, which predominate over tubes.

Establishing the physiological importance of IP6 in maturation is likely to be challenging. The capsid mutation R18A ablates HIV infectivity but does not prevent mature capsids from forming. Actually, they have been described as morphologically indistinguishable from their wild-type counterparts [29]. Moreover, R18A partially rescues the TRIM5 restriction of the wild-type virus during co-infection experiments, which provides functional evidence that it is correctly assembled. However, the mutant R18A virus fails to carry out any measurable DNA synthesis during infection, which suggests that it is unstable.

### 3.1. Post-Entry Infection

Post-entry infection occurs after fusion of the viral envelope with the target cell membrane and refers to the processes that occur between cytosolic deposition of the capsid and uncoating. Plausibly, the mature capsid has evolved to facilitate this step. It takes HIV-1 at least several hours to establish a productive infection of the cell and reverse transcription kinetics suggest that DNA synthesis peaks after approximately 8 to 10 h in cell lines [30,31,32]. In primary cells, reverse transcription kinetics may be even slower due to the limited pool of dNTPs resulting from SAMHD1 activity. Studies suggest a close relationship between SAMHD1, the viral antagonist Vpx, dNTP levels, and reverse transcription kinetics in macrophages [33]. Whether reverse transcription is completed before or after uncoating is unclear. Yet, it seems unlikely for uncoating to occur pre-strand transfer since this would dilute both the RT enzyme and the template RNA in the cytosol, which reduces catalytic efficiency. Moreover, early uncoating would risk exposing viral RNA and DNA to nucleic acid sensors such as RIG-I and cGAS, which can trigger a potent antiviral response [34]. Reverse transcription can occur in purified HIV-1 capsid cores where the enzyme and the template are at high local concentrations and viral nucleic acid is protected [23]. However, for the encapsidated RT enzyme to synthesize DNA, precursor dNTPs from the cytosol need to be able to access the capsid interior. In the initial description of the charged R18 hexamer ring and its ability to bind polyanions like dNTPs and hexacarboxybenzene, it was proposed that this feature might comprise a pore for nucleotide import [23]. However, structures of dNTP and ATP bound to HIV-1 capsid hexamers have shown that the interactions are driven by salt bridges, which do not provide a mechanism to discriminate between ribo-NTPs and the much less abundant deoxyribo-NTPs in the cytoplasm. If transport into the capsid is rate limiting, one might expect ATP to competitively inhibit dNTP entry. In fact, addition of ATP to isolated mature capsids increases reverse transcription rather than decreasing it [26]. It was postulated that ATP binding to the R18 pore was preventing uncoating, which suggests that polyanions might be used by the capsid as stabilizing cofactors.

IP6 greatly increases the thermostability of mature capsid hexamers by almost 10 °C [26]. Importantly, this stabilization is not due only to neutralization of the R18 charges since hexamers made with the R18G mutant protein are less stable than wildtype hexamers in the presence of IP6. These observations suggest that IP6 binding actually helps hold mature CA hexamers together. The estimated 300 IP6 molecules that are packaged per virion would be sufficient for each of the estimated 1800 CA molecules in the mature lattice to interact with IP6 [26]. Increased capsid stability may be of particular importance during reverse transcription. The addition of IP6 to isolated capsids potently increases viral DNA synthesis. Currently, no cellular data address the importance of IP6 in target cells while in vitro experiments suggest that IP6 stabilizes native HIV capsid cores on a physiologically meaningful time scale. Single molecule measurements have been made on individual GFP-containing HIV virions that are anchored to cover slips and permeabilized with a pore-forming toxin. In this assay, HIV-1 capsids have a lifetime of ~ 7 min, but, in the presence of IP6, they can remain intact and stable for more than 10 hours [26,35]. This result is particularly significant given the time it takes for HIV to reach the nucleus and complete RT. It suggests that, without IP6, the HIV-1 capsid is not stable for long enough to be infectious.

Which of the two possible functions for the R18 ring in post-entry HIV infection—nucleotide import and IP6-mediated capsid stabilization—is most important remains to be determined. The excellent correlation between loss of dNTP binding, reverse transcription, and infection when R18 was dose-dependently replaced by glycine initially suggested that dNTPs are imported through the pore to promote DNA synthesis [23]. However, as we now know that IP6 also binds to the R18 pore and increases the accumulation of viral DNA by stabilizing the capsid, this phenotype could be explained by a loss of IP6 interaction. A further complication is that it seems unlikely that each pore can accommodate more than one polyanion. Thus, a pore occupied by IP6 may be unable to act as a conduit for dNTPs. Moreover, IP6 is already present within the virion and coordinates the R18 ring more efficiently than dNTP and with likely a much slower off-rate. Nevertheless, there are hundreds of pores and it may be that import of dNTPs through only a few pores at a time is sufficient. Furthermore, if R18 pores are not involved in dNTP entry, then the question remains how nucleotides are imported to fuel DNA synthesis. It is possible that dNTPs slip between gaps in the capsid lattice, but tests of this idea currently are not available.

### 3.2. Uncoating

Exactly when (and where) uncoating occurs is controversial, but the capsid probably remains largely intact at least until the virus reaches the nuclear pore since interaction with the nuclear pore protein Nup153 requires mature hexamers [36]. At the same time, the capsid must be capable of rapid uncoating, which means that it must be structurally metastable. Using a small molecule cofactor like IP6 could enable HIV to switch its capsid from a very stable to a very unstable state simply by controlling IP6 binding. Remarkably, precedence for such a mechanism exists in virology. Picornaviruses use small molecule cofactors called pocket factors to regulate stability of their capsid [37]. These pocket factors are displaced during infection to facilitate uncoating [38].

If IP6 is an HIV-1 pocket factor, then how is its release controlled to trigger capsid uncoating? An attractive hypothesis is that cellular cofactors mediate uncoating, which could ensure that the virus uncoats in the right place at the right time and that capsid protection is lost only when it is no longer needed. For instance, uncoating may take place at the nuclear pore, which is the case for many nuclear DNA viruses such as adenoviruses [39] and herpes viruses [40]. Consistent with such an idea, the HIV-1 capsid interacts with several nuclear pore proteins and associated cofactors including Nup153, Nup358, and CPSF6 [36,41,42,43,44,45,46]. The relevant binding interfaces are localized close to each other in mature hexamers (Figure 3). It is plausible that allosteric pathways connect the various sites, which allows for a cofactor interaction to influence IP6 binding. One possible mechanism for how this might take place is if the cofactor interaction influences the orientation of the β-hairpin at the N-terminus of CA. When the β-hairpin adopts its closed conformation, IP6 is effectively trapped in an enclosed chamber inside the hexamer. However, when the β-hairpin is open, it becomes possible for IP6 to dissociate away from the capsid. There is, as of yet, no direct evidence for allostery in the HIV capsid but viral phenotypes are suggestive. For instance, CypA-binding loop mutants G89V and P90A are partially resistant to capsid inhibitor PF74 despite not impacting PF74 binding [47]. Capsid mutant E45A is also resistant to PF74 without preventing interaction and this residue is located proximal to the hydrogen bond network that determines the β-hairpin orientation. Moreover, E45A has a hyper-stable capsid phenotype, which could be explained by a model of increased β-hairpin closure and IP6 retention [48]. While we know that IP6 is packaged into virions before CA hexamer assembly and before the capsid is deposited into a target cell, it is not clear how efficiently IP6 or other polyanions can exchange in and out through the pore post-entry. Importantly, single molecule measurements suggest that, while hexacarboxybenzene and IP6 greatly stabilize the capsid, once uncoating has begun, the entire lattice quickly falls apart [35]. In contrast, while PF74 induces capsid opening—possibly by causing pentamer release [49]—it locks the lattice together. This suggests that a partially uncoated state is unlikely to persist since IP6 cannot be retained in a hexamer that has begun to lose subunits. It also indicates that, while IP6 loss may be the underlying mechanism that drives rapid uncoating, a small break in the capsid lattice or local conformational change may be sufficient to precipitate a chain-reaction. Thus, it may not be necessary to trigger the simultaneous release of IP6 across the entire capsid.

Another model that might explain loss of IP6 from the capsid is based on the ability of IP6 to strongly chelate divalent metals. The roughly millimolar concentration of Mg++ in the cytosol might lead to loss of IP6 due to its higher affinity for Mg++. Evaluation of this model would require the measurement of the affinity for the CA hexamer when assembled into a mature closed capsid.

## 4. Conclusions

Recent data suggest that IP6 has a key role in several stages of the HIV-1 lifecycle from assembly to uncoating. The following is a mechanistic working model for IP6 function. IP6 is recruited into assembling virions prior to budding where it promotes formation of the immature lattice by coordinating two rings of six lysine residues in Gag, K290 and K359. During maturation, viral protease severs CA from SP1, which destroys the immature IP6 binding site. Liberated IP6 drives assembly of mature hexamers by coordinating six R18 residues and facilitating formation of a metastable capsid structure. Upon infection, capsids are deposited into the cytosol where IP6 maintains capsid integrity during reverse transcription and transit to the nucleus. Environmental cues such as cofactor binding or DNA accumulation induce a rapid cascade of IP6 dissociation, which leads to lattice disassembly.

Many questions about IP6 function remain to be addressed for HIV-1 and more broadly for other retroviruses. Do other lentiviruses and other genera of retroviruses also use IP6 to modulate Gag and CA lattice stability? If not, how are their immature and mature lattices stabilized? Does IP6 play a role in replication of the many other viruses whose structures are based on protein hexamers? Do dNTPs enter the capsid through the R18 pore? Does IP6 in some way control the specificity for dNTPs? Does protein cofactor binding to the HIV-1 capsid lead to the release of IP6, as suggested by the model above? If so, what cofactor proteins play this role? The interplay of IP6 and viral structural proteins is likely to remain a fertile ground for investigation.

## Figures and Tables

**Figure 1 viruses-10-00640-f001:**
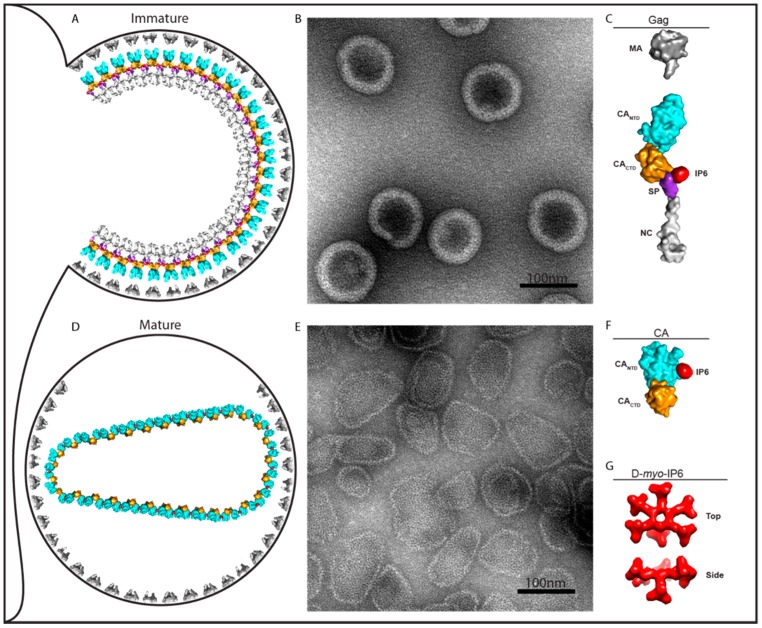
Role of IP6 on immature and mature assembly. (**A**) Drawing of an immature virus particle budding from cellular plasma membrane. (**B**) Negative stain EM image of immature-like virus particles assembled in vitro with IP6. (**C**) Depiction of Gag molecule from (**A**) showing IP6 bound at the interface of CA_CTD_ and SP. (**D**) Drawing of a mature virus particle. Following cleavage of Gag, IP6 promotes the formation of the mature lattice by CA. (**E**) Negative stain EM image of IP6-induced assembly of HIV-1 CA protein into mature-like virus particles. (**F**) Depiction of CA molecule from (**D**) with IP6 bound. (**G**) Top and side-view of IP6 *d*-myo-IP6, the most abundant isoform, with one axial and five equatorial phosphates. Note that in diagrams A and C, the viral envelope proteins and the RNA genome and associated proteins are not shown.

**Figure 2 viruses-10-00640-f002:**
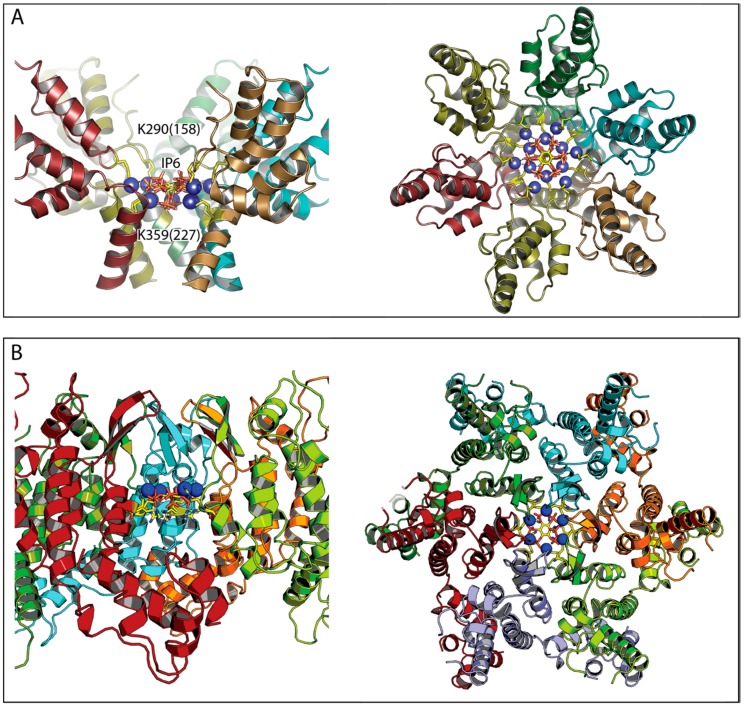
Immature and mature HIV capsid hexamers in complex with IP6. (**A**) Immature hexamer with location of the two lysine rings indicated by side-chains and blue spheres. Numbering is given for Gag with CA in parenthesis. Based on PDB 6BHR. The left-hand panel is annotated so that the lysine side chains belonging to K290 are immediately below the K290 label while those belonging to K359 are immediately above the K359 label. In the right-hand panel, the inner ring of blue spheres corresponds to lysines at position K359 while the outer ring of blue spheres corresponds to lysines at position K290. (**B**) Mature hexamer with R18 ring indicated by yellow side chains and blue spheres. Two R18 rotamer conformers are shown. Based on PDB 6ES8. In each side-view, one monomer has been omitted for clarity.

**Figure 3 viruses-10-00640-f003:**
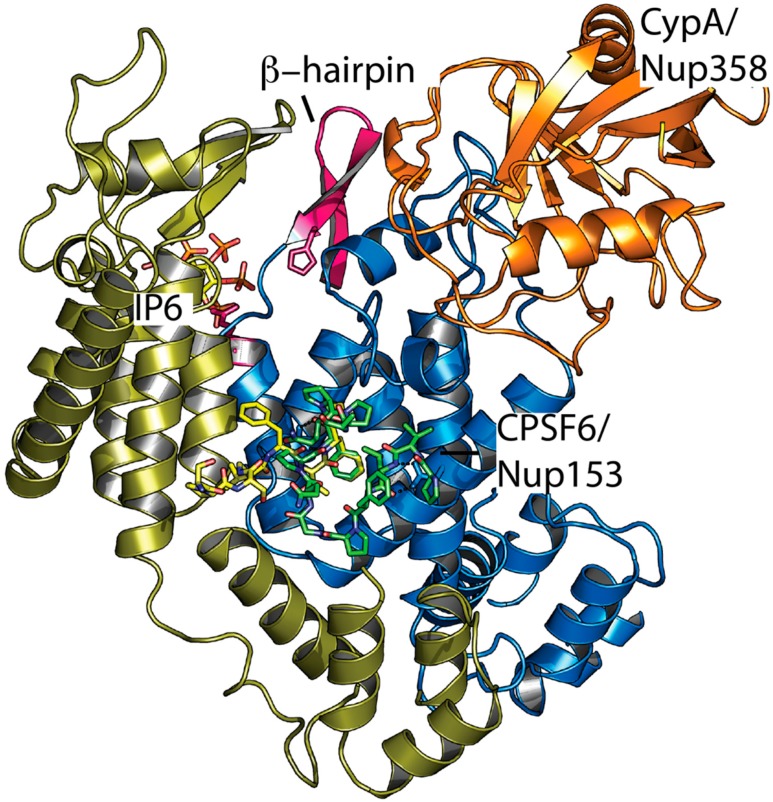
Ligand interaction sites on two adjacent monomers in the mature HIV-1 capsid hexamer. Superposition of complexed structures of CPSF6 (4U0B), Nup153 (4U0C), Nup358 (4LQW), and IP6 (6ES8). Two hexamer monomers are shown in blue and gold, CypA/Nup358 in orange, CPSF6 and Nup153 peptides in green and yellow, IP6 as atom colored sticks, and the β-hairpin in pink.

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
