# Peer review of "IP6 Regulation of HIV Capsid Assembly, Stability, and Uncoating"

_viruses, 2018, doi:10.3390/v10110640_

Reviewer 1 Report

The manuscript by Dick et al. discusses in a comprehensive, clear and exhaustive manner the role of inositol hexakisphosphate (IP6) in the assembly, stability and uncoating of the HIV-1 capsid. The authors review all the relevant literature on this subject, weighing its contribution in a fair an balanced way. Overall, this is an outstanding manuscript that will be of great help and important for those who work in this field of research.

Author Response

We are grateful for these comments and for the reviewers opinion that the review will be of interest to those in the field.  

Reviewer 2 Report

This is a very well written and timely review by authors who recently published major findings regarding the role of IP6 in HIV assembly and core formation. The authors summarize prior findings relevant to IP6 and then discuss work from their laboratories and others that have made this one of the most exciting findings of the past year in basic HIV virology. The figures provided in general are very helpful and nicely presented. The comments below include a suggestion for discussion that may help readers better synthesize the prior findings on IP6 and then some minor comments/corrections. Overall this is a very worthwhile summary of what is known about the role of IP6 and includes some intriguing speculative discussion of what its role might be in uncoating.

 Critique:

1.     The authors nicely summarize findings that led to the conclusion that IP6 interacts with MA and contribute to the elongated conformation of Gag in the second paragraph of page 3. They then summarize recent data that has clearly shown that MA is not required to promote immature assembly. Some additional discussion would be useful for clarity. Are the authors saying that there cannot be a role for an IP6-MA interaction in the cell that contributes to Gag elongation, or could this interaction and elongation precede the role of IP6 in stabilizing the 6-helix bundle of the CACTD-SP1 interface? Does interaction with the CACTD-SP1 interface itself lead to elongation of the Gag monomer, or is it only upon PI(4,5)P2 binding that Gag elongates, with IP6 playing no role in this conformational change?

 Minor:

1.     Figure 2: The labeling of key residues is not clear. In some panels such as the right side of 2A, a line points to the indicated residue. This is somewhat hard to see. In other panels it is not clear- is there a specific line indicating K290 or K359, or are the numbers intended to overly directly the residue indicated? Please make this more clear to the reader in each panel.

2.     Page 1 line 21 “polymerises” should be “polymerizes”

3.     Page 3, 3rd paragraph- this refers to “Dick et al.” in the third person, although Dick et al. are the authors. This may be a pretty minor point, but I’m used to seeing “We” to indicate that the current authors are discussing and reviewing their own work.

4.     Page 5 line 32 “RT reverse transcription”. The authors should use consistent terminology, and it seems most appropriate to spell out reverse transcription when the enzymatic process is referred to, and reserve RT for reference to the enzyme itself.

Author Response

Reviewer 1

This is a very well written and timely review by authors who recently published major findings regarding the role of IP6 in HIV assembly and core formation. The authors summarize prior findings relevant to IP6 and then discuss work from their laboratories and others that have made this one of the most exciting findings of the past year in basic HIV virology. The figures provided in general are very helpful and nicely presented. The comments below include a suggestion for discussion that may help readers better synthesize the prior findings on IP6 and then some minor comments/corrections. Overall this is a very worthwhile summary of what is known about the role of IP6 and includes some intriguing speculative discussion of what its role might be in uncoating.

 Critique:

Q1. The authors nicely summarize findings that led to the conclusion that IP6 interacts with MA and contribute to the elongated conformation of Gag in the second paragraph of page 3. They then summarize recent data that has clearly shown that MA is not required to promote immature assembly. Some additional discussion would be useful for clarity. Are the authors saying that there cannot be a role for an IP6-MA interaction in the cell that contributes to Gag elongation, or could this interaction and elongation precede the role of IP6 in stabilizing the 6-helix bundle of the CACTD-SP1 interface? Does interaction with the CACTD-SP1 interface itself lead to elongation of the Gag monomer, or is it only upon PI(4,5)P2 binding that Gag elongates, with IP6 playing no role in this conformational change?

 A1.

To clarify we’ve added the below sentence at the end of paragraph 3 of page 3 (line 26-27).

“Our findings do not rule out that IP6 interaction with MA and NC plays some role in Gag assembly.”

 Minor:

Q1. Figure 2: The labeling of key residues is not clear. In some panels such as the right side of 2A, a line points to the indicated residue. This is somewhat hard to see. In other panels it is not clear- is there a specific line indicating K290 or K359, or are the numbers intended to overly directly the residue indicated? Please make this more clear to the reader in each panel.

A1. To avoid any confusion, the labels have been omitted from most of the panels. Instead, description has been added to the legend: “The left-hand panel is annotated such that the lysine side chains belonging to K290 are immediately below the K290 label whilst those belonging to K359 are immediately above the K359 label. In the right-hand panel, the inner ring of blue spheres corresponds to lysines at position K359, whilst the outer ring of blue spheres corresponds to lysines at position K290.”

Q2.  Page 1 line 21 “polymerises” should be “polymerizes”

A2.  Corrected.

Q3. Page 3, 3rd paragraph- this refers to “Dick et al.” in the third person, although Dick et al. are the authors. This may be a pretty minor point, but I’m used to seeing “We” to indicate that the current authors are discussing and reviewing their own work.

A3. Given that this review covers recent work from two independent labs, referencing in the third person has been maintained for clarity of what ‘we’ means.

Q4. Page 5 line 32 “RT reverse transcription”. The authors should use consistent terminology, and it seems most appropriate to spell out reverse transcription when the enzymatic process is referred to, and reserve RT for reference to the enzyme itself.

A4. We have corrected the text such that RT refers to the enzyme and reverse transcription is spelt out when describing the process.

Reviewer 3 Report

The review entitled “IP6 regulation of HIV capsid assembly, stability and uncoating” regards an interesting topic, currently under investigation in several laboratories in the world. The review is well written and speculates on new researches in the HIV field. The authors found some elegant and interesting explanations about observed results, leaving often open questions for future investigations that could give rise in future a shared model on HIV uncoating. 

For example the two possible functions for the R18 ring in post-entry HIV infection – nucleotide import and IP6-mediated capsid stabilization. Therefore they also consider the possibility that R18 pores are not involved in dNTP entry, then the question remains, how nucleotides are imported to fuel DNA synthesis. A possibility that it is also raised by several groups as well as from the authors of this review  it is the possibility that dNTPs slip between gaps in the capsid lattice. Therefore there is no yet an answer for that.

Overall the review is of high interest for HIV field, therefore there are just few minor comments that should be addressed before publication.

Line 31: “a productive infection of the cell, and reverse transcription kinetics suggest that DNA synthesis peaks after approximately 8-10 hours.” 

It is important to specify if this data comes from endogenous RT in vitro or from  cells. The pick of RT is completely different if measured in T cells  or in macrophages, more details should be added.

Sometimes I found a weakness on citations, in fact some references have not been cited, which it is usually an important point for a review article. For example in the paragraph line 42 “HIV-1 capsid interacts with several nuclear pore proteins and associated cofactors, including Nu153, Nup358 and CPSF6[32, 37, 38]”

The authors cited only them self, however some previous studies found the interaction of RanBP2/ Nup358 with CA monomers (Schaller et al., Plos pathogens 2011) or with in vitro cores (Di Nunzio et al., Plos One 2012), as well as for Nup153 (Di Nunzio et al., Virology, 2013; Matreyek et al., Plos Pathogens, 2013).

In the sentence on the line 42 there is a  misspelling Nu153 instead then Nup153 

References are missed also in the following paragraph: “network that determines beta-hairpin orientation. Moreover, E45A has a hyperstable capsid phenotype”

The laboratory of Christophe Aiken has been the first to identify the aforementioned CA mutants but references are missed. 

Author Response

Reviewer 3

 The review entitled “IP6 regulation of HIV capsid assembly, stability and uncoating” regards an interesting topic, currently under investigation in several laboratories in the world. The review is well written and speculates on new researches in the HIV field. The authors found some elegant and interesting explanations about observed results, leaving often open questions for future investigations that could give rise in future a shared model on HIV uncoating. 

For example the two possible functions for the R18 ring in post-entry HIV infection – nucleotide import and IP6-mediated capsid stabilization. Therefore they also consider the possibility that R18 pores are not involved in dNTP entry, then the question remains, how nucleotides are imported to fuel DNA synthesis. A possibility that it is also raised by several groups as well as from the authors of this review  it is the possibility that dNTPs slip between gaps in the capsid lattice. Therefore there is no yet an answer for that. Overall the review is of high interest for HIV field, therefore there are just few minor comments that should be addressed before publication.

Q1. Line 31: “a productive infection of the cell, and reverse transcription kinetics suggest that DNA synthesis peaks after approximately 8-10 hours.” It is important to specify if this data comes from endogenous RT in vitro or from cells. The pick of RT is completely different if measured in T cells  or in macrophages, more details should be added.

A1. We have added additional references and text as follows: “It takes HIV-1 at least several hours to establish a productive infection of the cell, and reverse transcription kinetics suggest that DNA synthesis peaks after approximately 8-10 hours in cell lines[30-32]. In primary cells, reverse transcription kinetics may be even slower due to the limited pool of dNTPs resulting from SAMHD1 activity. Indeed, studies suggest a close relationship between SAMHD1, the viral antagonist Vpx, dNTP levels and reverse transcription kinetics in macrophages[33].”

Q2. Sometimes I found a weakness on citations, in fact some references have not been cited, which it is usually an important point for a review article. For example in the paragraph line 42 “HIV-1 capsid interacts with several nuclear pore proteins and associated cofactors, including Nu153, Nup358 and CPSF6[32, 37, 38]”

The authors cited only them self, however some previous studies found the interaction of RanBP2/ Nup358 with CA monomers (Schaller et al., Plos pathogens 2011) or with in vitro cores (Di Nunzio et al., Plos One 2012), as well as for Nup153 (Di Nunzio et al., Virology, 2013; Matreyek et al., Plos Pathogens, 2013).

A2. It is unfortunately not possible to cite all the papers dealing with HIV cofactors. As ours is a structural focus, we have cited those papers describing structural data. Nevertheless, we have added several of the suggested references. 

Q3.In the sentence on the line 42 there is a  misspelling Nu153 instead then Nup153 

A3. Corrected, thank you.

Q4. References are missed also in the following paragraph: “network that determines beta-hairpin orientation. Moreover, E45A has a hyperstable capsid phenotype”The laboratory of Christophe Aiken has been the first to identify the aforementioned CA mutants but references are missed. 

A4. A reference has been added to that sentence.